# On AUV Control with the Aid of Position Estimation Algorithms Based on Acoustic Seabed Sensing and DOA Measurements

**DOI:** 10.3390/s19245520

**Published:** 2019-12-13

**Authors:** Alexander Miller, Boris Miller, Gregory Miller

**Affiliations:** 1Institute for Information Transmission Problems RAS, 19/1 Bolshoy Karetny per., Moscow 127051, Russia; amiller@iitp.ru (A.M.); bmiller@iitp.ru (B.M.); 2Kazan Federal University, 18 Kremlyovskaya str., Kazan 420008, Russia; 3Monash University, Melbourne, VIC 3800, Australia; 4Institute of Informatics Problems of Federal Research Center “Computer Science and Control” RAS, 44/2 Vavilova str., Moscow 119333, Russia

**Keywords:** AUV navigation, position estimation, motion control, conditionally minimax nonlinear filtering, pseudo-measurements

## Abstract

This article discusses various approaches to the control of autonomous underwater vehicles (AUVs) with the aid of different velocity-position estimation algorithms. Traditionally this field is considered as the area of the extended Kalman filter (EKF) application: It became a universal tool for nonlinear observation models and its use is ubiquitous. Meanwhile, the specific characteristics of underwater navigation, such as an incomplete sets of measurements, constraints on the range metering or even impossibility of range measurements, observations provided by rather specific acoustic beacons, sonar observations, and other features seriously narrow the applicability of common instruments due to a high level of uncertainty and nonlinearity. The AUV navigation system, not being able to rely on a single source of position estimation, has to take into account all available information. This leads to the necessity of various complex estimation and data fusion algorithms, which are the matter of the present article. Here we discuss some approaches to the AUV position estimation such as conditionally minimax nonlinear filtering (CMNF) and unbiased pseudo-measurement filters (UPMFs) in conjunction with velocity estimation based on the seabed profile acoustic sensing. The presented estimation algorithms serve as a basis for a locally optimal AUV motion control algorithm, which is also presented.

## 1. Introduction

Navigation of AUVs (autonomous underwater vehicles), being essentially different from the navigation of UAVs (unmanned autonomous vehicles) due to the principal difference between the sources of navigation information, nevertheless shares several characteristic features with the latter. Indeed, in both cases, the INS (inertial navigation system) plays the main role in determining the position-velocity state of the vehicle. Navigation with the aid of INS is usually based on the dynamic model of the vehicle motion with inputs from various sensors of linear and angular velocities and uses filters (usually of Kalman type) for calculation of the position and attitude. However, all types of velocity sensors used in INS are subject to drift, which must be systematically compensated to keep the position estimation accuracy at an acceptable level. In UAV navigation such compensation may be accomplished with the aid of satellite positioning systems (GPS or GLONASS), or in the case of a so-called GPS denied environment, with the aid of specific video survey tools [1]. In an underwater environment, all electromagnetic (and therefore video tools) have a very restricted range of operation, and only acoustic waves permit sensing and communication for rather long distances. The problem is that acoustic measurements are notably barren in comparison with the electromagnetic ones: They do not have such features as color or contrast and just measure the direction of arrival (DOA) and/or the distance from the vehicle to the reflecting surface, such as obstacle, seabed, or ship. Therefore, even using multiple irradiating sensors like acoustic sonars, one can only obtain an image of distances to the seabed relief. An introduction of acoustical beacons with known positions can help to determine the own AUV position by using an angular observation filter, and this is the area of the (short and/or long) baseline methods application [2]. These methods are especially effective in the limited area of operation; moreover, they may be used in the case of a moving baseline established on a ship [3] and even in docking maneuvering [4]. Acoustic methods used in active mode give access to the distance measurements, which make the position estimation problem much easier, though in some areas of application such measurements are not acceptable or must be restricted due to the energetic or masking constraints [5].

The possibility of the sonar employment in AUV navigation has already been mentioned by some authors either in application to the situation, when an accurate seabed map is available, or in the case of presence of significant seabed features [6]. Moreover, the active pinging can be used to determine one’s speed over the seabed by tracking its features or by measuring the Doppler shift of returns [7], the same idea is used in the well-known DVL (Doppler velocity log) [8]. Moreover, acoustic sonars admit the usage of phased array technology, which permits to achieve the angular resolution up to 0.1° [9]. Angular measurements and particularly the bearing-only observations which are inherent to underwater acoustics lead to the necessity of special type algorithms for the position and velocity estimation in AUV navigation and tracking underwater targets [10]. All such algorithms are inevitably the modifications of the Kalman filter, especially in the problems dealing with the fusion of different sources of navigation information such as INS and DVL [11], INS and various geo fields, and INS and sonars [12,13]. In the present work, the approach to sonar terrain matching [14] will be investigated by analogy to the UAV optical flow. The principal difference from the standard Lucas-Kanade [15] algorithm is the presence of the AUV velocity components in observations only in implicit form, fortunately in a linear one, but the coefficients depend on the surface slope and must be estimated also via observation of the range profile. This means the problem is nonlinear and therefore the evaluation of the quality of the velocity estimation is possible via simulation only. Some preliminary results have already been obtained in [16,17]; below we discuss the extension of this approach.

The great majority of external navigation sensors provide nonlinear measurements of position- velocity parameters. That is the reason why in most cases the linear filtering theory does not apply. A common way to overcome this issue is the usage of various modifications of the Kalman filter, such as the extended Kalman filter (EKF) [18], particle filter (PF) [19], and unscented Kalman filter (UKF) [20]. However, careful analysis shows that all these filters, even if different from the viewpoint of complexity, give almost the same level of the AUV position estimation accuracy, especially in the case of bearing-only observations [21]. An alternative to traditional Kalman filtering is the conditionally optimal nonlinear filter proposed by V. S. Pugachev and developed by his successors into the conditionally minimax nonlinear filter (CMNF) [22]. This theory permits to develop the estimation procedure with given properties, for example, either with minimum dispersion or without bias, which is extremely important, for example, in target tracking based on bearing-only observations [23,24].

In acoustic position estimation and target tracking, the bearing-only observation plays the principal role. As it was observed many years ago, the bearing angles may be transformed into linear observation, but with bias [25]. This approach is called pseudo-measurements. In recent work, an unbiased modification of this method was proposed for UAV navigation based on radio beacons [26] and on a video recording of terrain objects with known coordinates [27].

In the present article, we consider new approaches to AUV navigation and control and compare them with traditional ones. Both the problems of control and position estimation are considered in the locally optimal formulation since the stochastic problems with nonlinear observations do not have explicit globally optimal solutions. The model is described in Section 2. In Section 3 the following problem of nominal trajectory is formulated as a control problem and its locally optimal solution is presented. Section 4 contains a position estimation algorithm based on dead reckoning with speed evaluated from the evolution of the “acoustic images”—the seabed distances registered with the aid of acoustic sonar. Section 5 is devoted to the filtering algorithms based on the dynamic model of the vehicle motion and DOA measurements. In Section 6 the performance of the locally optimal control based on the discussed filtering algorithms is evaluated through simulation.

## 2. The AUV Navigation System Model

Let the coordinates of the AUV at time instant $t_{k}$ be $X_{k}=X\left(t_{k}\right)={\left(X\left(t_{k}\right),Y\left(t_{k}\right),Z\left(t_{k}\right)\right)}^{T}={\left(X_{k},Y_{k},Z_{k}\right)}^{T}$, where ${\left(\cdot \right)}^{T}$ is the transposition operator. The sequence $t_{k}$, k=0,…,N, $t_{0}=0$, $t_{N}=T$ is known, and the interval between the consequent time instants is assumed to be the same for the sake of simplicity: $t_{k+1}-t_{k}=\Delta t$, k=0,…,N−1. The initial condition $X_{0}$ is a random vector with known expectation $m_{0}$ and covariance matrix $S_{0}$, the distribution $X_{0}∼P^{X_{0}}$ is assumed unknown. Denoting by P(m,S) the set of all distributions with expectation m and covariance S, we can write $P^{X_{0}}\in P\left(m_{0},S_{0}\right)$.

Let the controlled motion model of the AUV be defined by the following vector equation:(1)$$X_{k+1}=X_{k}+V_{k}\Delta t+W_{k},$$
where $V_{k}$ is the vector of the AUV speed at the moment $t_{k}$ and $W_{k}={\left(W_{k}^{X},W_{k}^{Y},W_{k}^{Z}\right)}^{T}$ is a sequence of independent and identically distributed (i.i.d.) random vectors with zero mean and covariance matrix $S^{W}$ independent from the initial condition: $W_{k}∼P^{W_{k}}\in P\left(0,S^{W}\right)$. The exact distribution of these vectors is assumed unknown.

The AUV speed vector is defined by its absolute value $V_{k}=∥V_{k}∥$ and two angles $\gamma_{k},\theta_{k}$ shown in Figure 1. The angle $\gamma_{k}$, is the angle between $V_{k}$ and the horizontal plane x0y, and $\theta_{k}$ is the angle between the projection of $V_{k}$ on the horizontal plane x0y and the axis 0x. Thus, the components of the speed vector are defined by $V_{k}=V_{k}{\left(\cos\gamma_{k}\cos\theta_{k},\cos\gamma_{k}\;\sin\theta_{k},\sin\gamma_{k}\right)}^{T}$ and vector model given by Equation (Equation 1) can be rewritten in the form of a system of scalar equations:(2)$$\left\{\begin{array}{c} X_{k+1}=X_{k}+V_{k}\cos\gamma_{k}\cos\theta_{k}\Delta t+W_{k}^{X},\\ Y_{k+1}=Y_{k}+V_{k}\cos\gamma_{k}\sin\theta_{k}\Delta t+W_{k}^{Y},\\ Z_{k+1}=Z_{k}+V_{k}\sin\gamma_{k}\Delta t+W_{k}^{Z}. \end{array}\right.$$

The vector of three parameters $u_{k}={\left(\gamma_{k},\theta_{k},V_{k}\right)}^{T}$ that define the direction and the absolute value of the speed vector $V_{k}=V_{k}\left(u_{k}\right)$ (and hence the whole dynamics of the AUV), is further called the control vector.

## 3. Locally Optimal AUV Path Control

We define the control problem for the AUV as a problem of following some predefined path ${\overset{˚}{X}}_{k}$, k=0,…,N, which is given by the control sequence ${\overset{˚}{u}}_{k}={\left({\overset{˚}{\gamma }}_{k},{\overset{˚}{\theta }}_{k},{\overset{˚}{V}}_{k}\right)}^{T}$ and calculated in the absence of any noise:(3)$$\left\{\begin{array}{c} {\overset{˚}{X}}_{k+1}={\overset{˚}{X}}_{k}+{\overset{˚}{V}}_{k}\cos{\overset{˚}{\gamma }}_{k}\cos{\overset{˚}{\theta }}_{k}\Delta t,\\ {\overset{˚}{Y}}_{k+1}={\overset{˚}{Y}}_{k}+{\overset{˚}{V}}_{k}\cos{\overset{˚}{\gamma }}_{k}\sin{\overset{˚}{\theta }}_{k}\Delta t,\\ {\overset{˚}{Z}}_{k+1}={\overset{˚}{Z}}_{k}+{\overset{˚}{V}}_{k}\sin{\overset{˚}{\gamma }}_{k}\Delta t. \end{array}\right.$$

We first consider the optimal control problem with full information, where the real coordinates of the AUV $X_{k}$ and its speed $V_{k}$ are known at all times $t_{k}$, k=0,…,N.

The quality of the control at time instant $t_{k}$ is defined by the following criterion:(4)$$J\left(u_{k}\right)=E\left\{{∥{\overset{˚}{X}}_{k+1}-X_{k+1}\left(u_{k}\right)∥}^{2}\right\}=E\left\{{\left({\overset{˚}{X}}_{k+1}-X_{k+1}\right)}^{2}+{\left({\overset{˚}{Y}}_{k+1}-Y_{k+1}\right)}^{2}+{\left({\overset{˚}{Z}}_{k+1}-Z_{k+1}\right)}^{2}\right\}$$

The control $u_{k}^{*}={\left(\gamma_{k}^{*},\theta_{k}^{*},V_{k}^{*}\right)}^{T}$ that minimizes Criterion (Equation 4) is called locally optimal since at any time instant $t_{k}$ it provides the least deviation from the nominal path on the next step $t_{k+1}$.

**Lemma** **1.**
*Let $\Delta {\overset{˚}{X}}_{k}=\left(\Delta {\overset{˚}{X}}_{k},\Delta {\overset{˚}{Y}}_{k},\Delta {\overset{˚}{Z}}_{k}\right)={\left({\overset{˚}{X}}_{k}-X_{k},{\overset{˚}{Y}}_{k}-Y_{k},{\overset{˚}{Z}}_{k}-Z_{k}\right)}^{T}$ denote the deviation of the real path of the AUV from the nominal one at the time instant $t_{k}$. Let also ${\overset{˚}{V}}_{k}={\left({\overset{˚}{V}}_{k}^{X},{\overset{˚}{V}}_{k}^{Y},{\overset{˚}{V}}_{k}^{Z}\right)}^{T}={\overset{˚}{V}}_{k}{\left(\cos{\overset{˚}{\gamma }}_{k}\cos{\overset{˚}{\theta }}_{k},\cos{\overset{˚}{\gamma }}_{k}\sin{\overset{˚}{\theta }}_{k},\sin{\overset{˚}{\gamma }}_{k}\right)}^{T}$ denote the nominal speed vector and its components. Then the solution to the optimization problem:*
(5)$$u_{k}^{*}=\underset{u_{k}}{\mathrm{argmin}}J\left(u_{k}\right)$$
*is given by the following relations:*
(6)$$\begin{array}{c} \theta_{k}^{*}=\mathrm{atan}2\left(\Delta {\overset{˚}{Y}}_{k}+\Delta t{\overset{˚}{V}}_{k}^{Y},\;\Delta {\overset{˚}{X}}_{k}+\Delta t{\overset{˚}{V}}_{k}^{X}\right),\\ \gamma_{k}^{*}=\mathrm{atan}2\left(\Delta {\overset{˚}{Z}}_{k}+\Delta t{\overset{˚}{V}}_{k}^{Z},\;\left(\Delta {\overset{˚}{X}}_{k}+\Delta t{\overset{˚}{V}}_{k}^{X}\right)/\cos\theta_{k}^{*}\right),\\ V_{k}^{*}=∥\Delta {\overset{˚}{X}}_{k}+\Delta t{\overset{˚}{V}}_{k}∥/\Delta t, \end{array}$$
*where 2(y,x) is the angle between the positive direction of axis 0x and the vector ${\left(x,y\right)}^{T}$.*


The proof of Lemma 1 is given in Appendix B.

The relations for the optimal angles and the absolute value have straightforward geometrical interpretation. From Figure 2, it follows that the locally optimal control at the time instant $t_{k}$ determines the AUV movement speed $V_{k}^{*}=V_{k}^{*}{\left(\cos\gamma_{k}^{*}\cos\theta_{k}^{*},\;\cos\gamma_{k}^{*}\sin\theta_{k}^{*},\sin\gamma_{k}^{*}\right)}^{T}$ aiming towards the point of the nominal path ${\overset{˚}{X}}_{k+1}$ and possessing the magnitude $V_{k}^{*}$, which is necessary to reach that point at the next time instant $t_{k+1}$.

The last observation provides the justification for a control policy, which is derived from Relations (Equation 6) by substituting the real AUV coordinates $X_{k}$ by an unbiased estimate ${\hat{X}}_{k}$: (7)$$\begin{array}{c} {\hat{\theta }}_{k}^{*}=2\left({\overset{˚}{Y}}_{k}-{\hat{Y}}_{k}+\Delta t{\overset{˚}{V}}_{k}^{Y},\;{\overset{˚}{X}}_{k}-{\hat{X}}_{k}+\Delta t{\overset{˚}{V}}_{k}^{X}\right),\\ {\hat{\gamma }}_{k}^{*}=2\left({\overset{˚}{Z}}_{k}-{\hat{Z}}_{k}+\Delta t{\overset{˚}{V}}_{k}^{Z},\;\left({\overset{˚}{X}}_{k}-{\hat{X}}_{k}+\Delta t{\overset{˚}{V}}_{k}^{X}\right)/\cos\theta_{k}^{*}\right),\\ {\hat{V}}_{k}^{*}=∥{\overset{˚}{X}}_{k}-{\hat{X}}_{k}+\Delta t{\overset{˚}{V}}_{k}∥/\Delta t \end{array}$$

The original optimal control problem for the nonlinear stochastic System (Equation 2) with incomplete information is rather hard for analysis. Moreover, it does not have an explicit solution even in the case of the quadratic criterion. In these circumstances, suboptimal control policies like Relations (Equation 7) seems reasonable substitution for the unattainable optimum.

Another feature of the optimal control in Relations (Equation 6) is that in general, it is unbounded, whereas for the real AUV navigation system the limits on the instantaneous course and speed change (or acceleration constraints) would be natural. Nevertheless, this does not affect the applicability of the estimation methods, which are the primary focus of this study. In presence of such constraints, optimization Problem (Equation 5) no longer has a solution in an explicit form, but since it is a well-known quadratic optimization problem with linear (box) constraints, the solution can be calculated using numerous effective numerical methods [28].

## 4. Position Estimation with Seabed Sensing

In this section, we present an approach to the position estimation based on dead reckoning with speed evaluated from the evolution of the “acoustic images”. Each image consists of the seabed distance measurements $L_{k}^{ij}=L\left(X_{k},u_{k},\gamma^{i},\theta^{j}\right)$ made at the same time instant $t_{k}$ by acoustic sensors aimed at different angles $\left(\gamma^{i},\theta^{j}\right)$ with respect to the AUV attitude, given by its movement direction $u_{k}$. The information about the AUV position shift $\Delta X_{k+1}=X_{k+1}-X_{k}$, or the speed $V_{k}$ is then derived from the difference between the values of correspondent distance measurements $L_{k}^{ij}$ and $L_{k+1}^{ij}$ made at the consecutive time instants $t_{k+1}$ and $t_{k}$. The way this is done is close to the Lucas-Kanade method for optical flow estimation [15] with the set of acoustic measurements treated as an image and a single distance to the seabed treated as a pixel intensity.

In detail, the model of acoustic measurements is presented in Figure 3. Let $\left(\gamma^{i},\theta^{j}\right)$, i,j=1,…,M be the set of aiming angles at which the sensors of the AUV emit the acoustic beams and acquire the distance to the seabed. These angles are defined with respect to the AUV attitude, which means that the absolute beam (i,j) direction is given by the angles $\left(\gamma_{k}+\gamma^{i},\theta_{k}+\theta^{j}\right)$. Thus the coordinates of the point $x_{k}^{ij}={\left(x_{k}^{ij},y_{k}^{ij},z_{k}^{ij}\right)}^{T}$, where the beam (i,j) reaches the seabed at the time instant $t_{k}$ are expressed as:(8)$$x_{k}^{ij}=X_{k}+L_{k}^{ij}e_{k},$$
where $e_{k}={\left(e_{k}^{X},e_{k}^{Y},e_{k}^{Z}\right)}^{T}={\left(\cos\left(\gamma_{k}+\gamma^{i}\right)\cos\left(\theta_{k}+\theta^{j}\right),\cos\left(\gamma_{k}+\gamma^{i}\right)\sin\left(\theta_{k}+\theta^{j}\right),\sin\left(\gamma_{k}+\gamma^{i}\right)\right)}^{T}$ defines the direction of the beam. It should be noted that this model differs from the one considered in [16,17] since it takes into account the controlled AUV attitude, whether in the mentioned works the beam directions were assumed constant. Note also, that the number of aiming angles $\gamma^{i}$ and $\theta^{j}$ could be assumed different with no effect on the further derivations.

Let the seabed profile be defined by the equation ψ(x)=0, where ψ(·) is some smooth function. Then considering x as a function of variables X, *L*, and e, each of which in turn depends on the time *t*:x(t)=X(t)+L(t)e(t),
and calculating the total derivative $\frac{d\psi (x)}{dt}=\frac{d\psi (x(X(t),L(t),e(t)))}{dt}$, we get:(9)$$\frac{\delta \psi }{\delta x}\left(\frac{dX}{dt}+\frac{de^{X}}{dt}L+e^{X}\frac{dL}{dt}\right)+\frac{\delta \psi }{\delta y}\left(\frac{dY}{dt}+\frac{de^{Y}}{dt}L+e^{Y}\frac{dL}{dt}\right)+\frac{\delta \psi }{\delta z}\left(\frac{dZ}{dt}+\frac{de^{Z}}{dt}L+e^{Z}\frac{dL}{dt}\right)=0,$$
where the actual functions’ arguments were omitted for the sake of simplicity. Assume that the partial derivatives of the function ψ(·) are known at any point of the seabed which could be reached with an acoustic beam. Rewriting Equation (Equation 9) in discrete time with substitution of the differentials with corresponding increments:
(10)$$\begin{array}{ccc} \Delta X_{k+1} & = & X_{k+1}-X_{k}={\left(\Delta X_{k+1},\Delta Y_{k+1},\Delta Z_{k+1}\right)}^{T},\\ \Delta L_{k+1}^{ij} & = & L\left(X_{k+1},u_{k+1},\gamma^{i},\theta^{j}\right)-L\left(X_{k},u_{k},\gamma^{i},\theta^{j}\right),\\ \Delta e_{k+1} & = & {\left(\Delta e_{k+1}^{X},\Delta e_{k+1}^{Y},\Delta e_{k+1}^{Z}\right)}^{T}\\ \Delta e_{k+1}^{X} & = & -\Delta \gamma_{k+1}\left(\sin\left(\gamma_{k}+\gamma^{i}\right)\cos\left(\theta_{k}+\theta^{j}\right)\right)-\Delta \theta_{k+1}\left(\cos\left(\gamma_{k}+\gamma^{i}\right)\sin\left(\theta_{k}+\theta^{j}\right)\right),\\ \Delta e_{k+1}^{Y} & = & -\Delta \gamma_{k+1}\left(\sin\left(\gamma_{k}+\gamma^{i}\right)\sin\left(\theta_{k}+\theta^{j}\right)\right)-\Delta \theta_{k+1}\left(\cos\left(\gamma_{k}+\gamma^{i}\right)\cos\left(\theta_{k}+\theta^{j}\right)\right),\\ \Delta e_{k+1}^{Z} & = & \Delta \gamma_{k+1}\left(\cos(\gamma_{k}+\gamma^{i})\right),\\ \Delta \gamma_{k+1} & = & \gamma_{k+1}-\gamma_{k},\\ \Delta \theta_{k+1} & = & \theta_{k+1}-\theta_{k}, \end{array}$$
we have the following equation:$$\begin{array}{ccc} \frac{\delta \psi }{\delta x}\left(x_{k}^{ij}\right)\left(\Delta X_{k+1}+\Delta e_{k+1}^{X}L_{k}^{ij}+e_{k}^{X}\Delta L_{k+1}^{ij}\right)\\ +\frac{\delta \psi }{\delta y}\left(x_{k}^{ij}\right)\left(\Delta Y_{k+1}+\Delta e_{k+1}^{Y}L_{k}^{ij}+e_{k}^{Y}\Delta L_{k+1}^{ij}\right)\\ +\frac{\delta \psi }{\delta z}\left(x_{k}^{ij}\right)\left(\Delta Z_{k+1}+\Delta e_{k+1}^{Z}L_{k}^{ij}+e_{k}^{Z}\Delta L_{k+1}^{ij}\right) & = & 0, \end{array}$$
which in turn can be expressed with unknowns $\Delta X_{k+1}$, $\Delta Y_{k+1}$, $\Delta Z_{k+1}$ collected on the left-hand side:(11)$$\frac{\delta \psi }{\delta x}\left(x_{k}^{ij}\right)\Delta X_{k+1}+\frac{\delta \psi }{\delta y}\left(x_{k}^{ij}\right)\Delta Y_{k+1}+\frac{\delta \psi }{\delta z}\left(x_{k}^{ij}\right)\Delta Z_{k+1}=B_{k}^{ij},$$
with
(12)$$\begin{array}{ccc} B_{k}^{ij} & = & -\frac{\delta \psi }{\delta x}\left(x_{k}^{ij}\right)\left(\Delta e_{k+1}^{X}L_{k}^{ij}+e_{k}^{X}\Delta L_{k+1}^{ij}\right)\\ & & -\frac{\delta \psi }{\delta y}\left(x_{k}^{ij}\right)\left(\Delta e_{k+1}^{Y}L_{k}^{ij}+e_{k}^{Y}\Delta L_{k+1}^{ij}\right)\\ & & -\frac{\delta \psi }{\delta z}\left(x_{k}^{ij}\right)\left(\Delta e_{k+1}^{Z}L_{k}^{ij}+e_{k}^{Z}\Delta L_{k+1}^{ij}\right). \end{array}$$

The unknowns $\Delta X_{k+1}$, $\Delta Y_{k+1}$, $\Delta Z_{k+1}$ can not be derived from a single equation, but assuming that these shifts are the same for all the acoustic sensors, we have a set of Equation (Equation 11) for different aiming angle values $\left(\gamma^{i},\theta^{j}\right)$, i,j=1,…,M. Finally it is possible to estimate the unknowns with the least-squares method:(13)$$\Delta {\hat{X}}_{k+1}=\underset{\Delta X_{k+1}}{\mathrm{argmin}}\sum_{i,j=1}^{M}{\left(\frac{\delta \psi }{\delta x}\left(x_{k}^{ij}\right)\Delta X_{k+1}+\frac{\delta \psi }{\delta y}\left(x_{k}^{ij}\right)\Delta Y_{k+1}+\frac{\delta \psi }{\delta z}\left(x_{k}^{ij}\right)\Delta Z_{k+1}-B_{k}^{ij}\right)}^{2}$$

Vectorization of Equation (Equation 11) with respect to the set of aiming angles $\left(\gamma^{i},\theta^{j}\right)$, i,j=1,…,M gives:$$A_{k}\Delta X_{k+1}=B_{k},$$
where $A_{k}$ and $B_{k}$ are made of vertically stacked row-vectors and values corresponding to the individual observations:(14)$$A_{k}=\left(\begin{array}{ccc} \frac{\delta \psi }{\delta x}\left(x_{k}^{11}\right) & \frac{\delta \psi }{\delta x}\left(x_{k}^{11}\right) & \frac{\delta \psi }{\delta x}\left(x_{k}^{11}\right)\\ \vdots & \vdots & \vdots \\ \frac{\delta \psi }{\delta x}\left(x_{k}^{1M}\right) & \frac{\delta \psi }{\delta x}\left(x_{k}^{1M}\right) & \frac{\delta \psi }{\delta x}\left(x_{k}^{1M}\right)\\ \vdots & \vdots & \vdots \\ \frac{\delta \psi }{\delta x}\left(x_{k}^{MM}\right) & \frac{\delta \psi }{\delta x}\left(x_{k}^{MM}\right) & \frac{\delta \psi }{\delta x}\left(x_{k}^{MM}\right) \end{array}\right),\;B_{k}=\left(\begin{array}{c} B_{k}^{11}\\ \vdots \\ B_{k}^{1M}\\ \vdots \\ B_{k}^{MM} \end{array}\right).$$

Now the solution to the least-squares optimization Problem (Equation 13) can be expressed in the standard form:(15)$$\Delta {\hat{X}}_{k+1}={\left[A_{k}^{T}A_{k}\right]}^{-1}A_{k}^{T}B_{k}.$$

Finally the AUV position dead reckoning estimate is:(16)$${\hat{X}}_{k+1}={\hat{X}}_{k}+\Delta {\hat{X}}_{k+1}.$$

It should be noted that Equation (Equation 16) requires the values of the partial derivatives of the function ψ(·) at the points $x_{k}^{ij}$, k=0,…,N, i,j=1,…,M, where the acoustic beams reach the seabed. Precise calculation of these values even in the case of known seabed profile is not possible since it requires the precise values of the points $x_{k}^{ij}$, which depend on the unavailable precise AUV position $X_{k}$ due to Equation (Equation 8). The way to overcome this issue is to use the acoustic image data to estimate the seabed slope, i.e., the required partial derivatives of ψ(·). In Appendix A, we present a method of estimation, based on the approximation of the slope function ψ(·), which requires only the values of seabed distances $L_{k}^{ij}$. Another method is proposed in [16].

The following algorithm summarizes the proposed acoustic seabed sensing AUV position estimation method:at time instant $t_{k+1}$ collect the seabed distance measurements $L_{k+1}^{ij}$ from the acoustic sensors i,j=1,…,M and calculate the increments $\Delta L_{k+1}^{ij}=L_{k+1}^{ij}-L_{k}^{ij}$;using the control values on the current $\left(\gamma_{k+1},\theta_{k+1}\right)$ and the previous $\left(\gamma_{k},\theta_{k}\right)$ steps calculate the increments ${\left(\Delta e_{k+1}^{X},\Delta e_{k+1}^{Y},\Delta e_{k+1}^{Z}\right)}^{T}$ according to Equation (Equation 10);evaluate the slope estimates $\hat{\frac{\delta \psi }{\delta x}}\left(x_{k}^{ij}\right)$, $\hat{\frac{\delta \psi }{\delta y}}\left(x_{k}^{ij}\right)$, $\hat{\frac{\delta \psi }{\delta z}}\left(x_{k}^{ij}\right)$ using Equation (Equation 27) or another method;using Equations (Equation 12) and (Equation 14) calculate the matrix $A_{k}$ and the vector $B_{k}$;calculate the estimate of the AUV position shift $\Delta {\hat{X}}_{k+1}$ with Equation (Equation 15) and the position estimate ${\hat{X}}_{k+1}$ with Equation (Equation 16).

## 5. Position Estimation with DOA Measurements

In this section we discuss various position estimation algorithms based on the dynamic model of the vehicle motion, Equation (Equation 1), and the external bearing-only measurements provided by a passive acoustic DOA estimation device (pressure hydrophone or acoustic vector sensor array) [6].

The position of the acoustic source $X_{B}={\left(X_{B},Y_{B},Z_{B}\right)}^{T}$ is assumed to be known and constant which is the case of a pre-deployed stationary acoustic beacon. It is assumed that at any time instant $t_{k}$ the bearing vector (Figure 4) is available for observation in the following form:(17)$$Y_{k}=\left(\begin{array}{c} \tan\varphi_{k}\\ \tan\lambda_{k} \end{array}\right),\;\begin{array}{c} \tan\varphi_{k}=\frac{Y_{B}-Y_{k}}{X_{B}-X_{k}}+\varepsilon_{k}^{\varphi },\\ \tan\lambda_{k}=\frac{\left(Z_{B}-Z_{k}\right)\cos\varphi_{k}}{X_{B}-X_{k}}+\varepsilon_{k}^{\lambda }, \end{array}$$
where $E_{k}={\left(\varepsilon_{k}^{\varphi },\varepsilon_{k}^{\lambda }\right)}^{T}∼P^{E_{k}}\in P\left(0,S^{\varepsilon }\right)$ is a sequence of i.i.d. random vectors independent from $W_{k}$ and $X_{0}$. The exact distribution of these vectors, as in the case of the noise in the Model (Equation 2), is assumed unknown.

Systems (Equation 2) and (Equation 17) can be expressed in the following general vector form:(18)$$\begin{array}{c} X_{k+1}=\Phi_{k}\left(X_{k},u_{k}\right)+W_{k},\\ Y_{k}=\Psi_{k}\left(X_{k}\right)+E_{k}. \end{array}$$

The common way to deal with the nonlinear System (Equation 18) is the extended Kalman filter. In this section we present alternative approaches, which in some cases allow to achieve better estimation quality then by direct system linearization. In Section 5.1, we present an observation Equation (Equation 17) transformation, allowing to reduce the original problem to a form, where the optimal filtering solution is also available in the form of the Kalman filter. In Section 5.2, we provide the conditionally minimax filtering problem statement and solution for the original nonlinear model of Equation (Equation 18) and show how this particular filtering approach allows data fusion from the DOA measurements and the dead reckoning navigation system based on the acoustic seabed sensing from Section 4.

### 5.1. Pseudo Measurements Filter

Rewriting the observations of Equation (Equation 17) in the following form:$$\begin{array}{c} \left(X_{B}-X_{k}\right)\sin\varphi_{k}=\left(Y_{B}-Y_{k}\right)\cos\varphi_{k}+\varepsilon_{k}^{\varphi }\left(X_{B}-X_{k}\right)\cos\varphi_{k},\\ \left(X_{B}-X_{k}\right)\sin\lambda_{k}=\left(Z_{B}-Z_{k}\right)\cos\varphi_{k}\cos\lambda_{k}+\varepsilon_{k}^{\lambda }\left(X_{B}-X_{k}\right)\cos\lambda_{k}, \end{array}$$
and gathering all the known or measured values at the left-hand side, we get:(19)$$\begin{array}{c} X_{B}\sin\varphi_{k}-Y_{B}\cos\varphi_{k}=X_{k}\sin\varphi_{k}-Y_{k}\cos\varphi_{k}+\varepsilon_{k}^{\varphi }\left(X_{B}-X_{k}\right)\cos\varphi_{k},\\ X_{B}\sin\lambda_{k}-Z_{B}\cos\varphi_{k}\cos\lambda_{k}=X_{k}\sin\lambda_{k}-Z_{k}\cos\varphi_{k}\cos\lambda_{k}+\varepsilon_{k}^{\lambda }\left(X_{B}-X_{k}\right)\cos\lambda_{k}. \end{array}$$

The right-hand side in the previous expression is linear with respect to the system state $X_{k}={\left(X_{k},Y_{k},Z_{k}\right)}^{T}$, but at the same time, it involves the additive noise, whose covariance is now state-dependent. Denote:$$Y_{k}^{\prime }=\left(\begin{array}{c} X_{B}\sin\varphi_{k}-Y_{B}\cos\varphi_{k}\\ X_{B}\sin\lambda_{k}-Z_{B}\cos\varphi_{k}\cos\lambda_{k} \end{array}\right)$$
and rewrite Equation (Equation 19) in the vector form:(20)$$Y_{k}^{\prime }=\Psi_{k}^{1}X_{k}+\Psi_{k}^{2}E_{k},$$
where
$$\begin{array}{c} \Psi_{k}^{1}=\Psi_{k}^{1}\left(Y_{k}\right)=\left(\begin{array}{ccc} \sin\varphi_{k} & -\cos\varphi_{k} & 0\\ \sin\lambda_{k} & 0 & -\cos\varphi_{k}\cos\lambda_{k} \end{array}\right)\\ \Psi_{k}^{2}=\Psi_{k}^{2}\left(X_{k},Y_{k}\right)=\left(\begin{array}{cc} \left(X_{B}-X_{k}\right)\cos\varphi_{k} & 0\\ 0 & \left(X_{B}-X_{k}\right)\cos\lambda_{k} \end{array}\right) \end{array}$$

The introduction of $Y_{k}^{\prime }$, which is called pseudo-measurements, is a common method of nonlinear systems analysis [21,25]. The idea is based on the fact that linear Kalman filtering estimate is the linear-optimal one in the case of linear observations and system dynamics and on the assumption that the noise $\Psi_{k}^{2}E_{k}$ covariance, even being state-dependent, may be evaluated or replaced by an upper bound. In our previous work [26] the filtering algorithm based on the pseudo-measurements has been suggested and its recurrence relations has been derived from the assumption of the unbiasedness of the estimate on the previous step. As applied to the problem at hand these relations are as follows:(21)$$\begin{array}{c} {\tilde{X}}_{k}=\Phi_{k-1}\left({\hat{X}}_{k-1},u_{k-1}\right),\\ {\tilde{K}}_{k}={\hat{K}}_{k-1}+S^{W},\\ {\bar{K}}_{k}={\tilde{K}}_{k}{\Psi_{k}^{1}}^{T}\left(Y_{k}\right){\left(\Psi_{k}^{1}\left(Y_{k}\right){\tilde{K}}_{k}{\Psi_{k}^{1}}^{T}\left(Y_{k}\right)+\Psi_{k}^{2}\left({\tilde{X}}_{k},Y_{k}\right)S^{\varepsilon }{\Psi_{k}^{2}}^{T}\left({\tilde{X}}_{k},Y_{k}\right)\right)}^{+}\\ {\hat{X}}_{k}={\tilde{X}}_{k}+{\bar{K}}_{k}\left(Y_{k}^{\prime }-\Psi_{k}^{1}X_{k}\right)\\ {\hat{K}}_{k}=\left(I-{\bar{K}}_{k}\Psi_{k}^{1}\left(Y_{k}\right)\right){\tilde{K}}_{k} \end{array}$$

This filter has the common Kalman structure with items recurrently calculated upon the bearing measurements update. The difference with the classic case is that here the Riccati equation is not solvable a priori, because it involves estimate-dependent terms and current measurement values.

### 5.2. Conditionnaly Minimax Nonlinear Filter

Let the functions $\Phi_{k}\left(\cdot ,\cdot \right)$, $\Psi_{k}\left(\cdot \right)$ and the feedback control $u_{k}=u_{k}\left(Y_{0},\ldots ,Y_{k}\right)$ be such that the first and second moments of the state $X_{k}$ and observation $Y_{k}$ are finite. Introduce two sets of functions: Basic prediction $\alpha_{k}\left(x,u\right)$ and basic correction functions $\beta_{k}\left(x,y\right)$. Then the CMNF estimate is defined by the following recurrent relations:(22)$$\begin{array}{cc} {\tilde{X}}_{k}=F_{k}\alpha_{k}\left({\hat{X}}_{k-1},u_{k-1}\right)+f_{k}, & F_{k}=\mathrm{cov}\left(X_{k},\alpha_{k}\left({\hat{X}}_{k-1},u_{k-1}\right)\right){\mathrm{cov}}^{+}\left(\alpha_{k}\left({\hat{X}}_{k-1},u_{k-1}\right),\alpha_{k}\left({\hat{X}}_{k-1},u_{k-1}\right)\right),\\ & f_{k}=E\left\{X_{k}\right\}-F_{k}E\left\{\alpha_{k}\left({\hat{X}}_{k-1},u_{k-1}\right)\right\}\\ {\hat{X}}_{k}={\tilde{X}}_{k}+H_{k}\beta_{k}\left({\tilde{X}}_{k},Y_{k}\right)+h_{k}, & H_{k}=\mathrm{cov}\left(X_{k}-{\tilde{X}}_{k},\beta_{k}\left({\tilde{X}}_{k},Y_{k}\right)\right){\mathrm{cov}}^{+}\left(\beta_{k}\left({\tilde{X}}_{k},Y_{k}\right),\beta_{k}\left({\tilde{X}}_{k},Y_{k}\right)\right),\\ & h_{k}=-H_{k}E\left\{\beta_{k}\left({\tilde{X}}_{k},Y_{k}\right)\right\}, \end{array}$$
where cov(x,y) is the covariance matrix of two random vectors x, y, $A^{+}$ denotes matrix pseudo-inversion, and ${\hat{X}}_{0}=m_{0}$.

In the case when the functions $\alpha_{k}\left(\cdot ,\cdot \right)$ and $\beta_{k}\left(\cdot ,\cdot \right)$ have first and second order moments for all the random arguments in Equation (Equation 22), then the CMNF estimate exists and has the following minimax property. Let ${\hat{X}}_{k-1}$ be the CMNF estimate at the time instant $t_{k-1}$. Then the linear functions $F_{k}^{*}\left(\xi \right)=F_{k}\xi +f_{k}$, $H_{k}^{*}\left(\zeta \right)=H_{k}\zeta +h_{k}$ defined by Equation (Equation 22) provide the solution for the following minimax optimization problem:(23)$$\begin{array}{c} F_{k}^{*}\left(\cdot \right)=\underset{F_{k}\left(\cdot \right)}{\mathrm{argmin}}\max_{P_{k}^{\prime }}E\left\{∥F_{k}\left(\alpha_{k}\left({\hat{X}}_{k-1},u_{k-1}\right)\right)-X_{k}{∥}^{2}\right\},\\ H_{k}^{*}\left(\cdot \right)=\underset{H_{k}\left(\cdot \right)}{\mathrm{argmin}}\max_{P_{k}^{\prime \prime }}E\left\{∥H_{k}\left(\beta_{k}\left({\tilde{X}}_{k},Y_{k}\right)\right)-\left(X_{k}-{\tilde{X}}_{k}\right){∥}^{2}\right\}, \end{array}$$
where $P_{k}^{\prime }\in P\left(E\left\{Z_{k}^{\prime }\right\},\mathrm{cov}\left(Z_{k}^{\prime },Z_{k}^{\prime }\right)\right)$—the set of all possible distributions of the compound vector $Z_{k}^{\prime }={\left(X_{k},\alpha_{k}\left({\hat{X}}_{k-1},u_{k-1}\right)\right)}^{T}$ and $P_{k}^{\prime \prime }\in P\left(E\left\{Z_{k}^{\prime \prime }\right\},\;\mathrm{cov}\left(Z_{k}^{\prime \prime },Z_{k}^{\prime \prime }\right)\right)$—the set of all possible distributions of the compound vector $Z_{k}^{\prime \prime }={\left(X_{k}-{\tilde{X}}_{k},\beta_{k}\left({\tilde{X}}_{k},Y_{k}\right)\right)}^{T}$.

The details on the CMNF approach to the nonlinear stochastic systems state estimation, including the thorough justification of Equation (Equation 22) being the solution to Equation (Equation 23) and the conditions of the solution in Equation (Equation 23) existence, could be found in [22]. Further application of the concept along with the comparative numerical study is the matter of the works [23,24].

The fact that the CMNF estimate, Equation (Equation 22), is the solution to the minimax problem, Equation (Equation 23), means that at each time instant $t_{k}$ it delivers the minimum for the worst case (with respect to the a priori uncertainty in the distributions $P_{k}^{\prime }$ and $P_{k}^{\prime \prime }$) of the mean-square error of the prediction ${\tilde{X}}_{k}$ and correction ${\hat{X}}_{k}$. It should be noted that both ${\tilde{X}}_{k}$ and ${\hat{X}}_{k}$ are unbiased estimates of $X_{k}$ and the quality of these estimates is a priori known:$$\begin{array}{c} \mathrm{cov}\left(X_{k}-{\tilde{X}}_{k},X_{k}-{\tilde{X}}_{k}\right)=\mathrm{cov}\left(X_{k},X_{k}\right)-F_{k}\mathrm{cov}\left(\alpha_{k}\left({\hat{X}}_{k-1},u_{k-1}\right),X_{k}\right),\\ \mathrm{cov}\left(X_{k}-{\hat{X}}_{k},X_{k}-{\hat{X}}_{k}\right)=\mathrm{cov}\left(X_{k}-{\tilde{X}}_{k},X_{k}-{\tilde{X}}_{k}\right)-H_{k}\mathrm{cov}\left(\beta_{k}\left({\tilde{X}}_{k},Y_{k}\right),X_{k}-{\tilde{X}}_{k}\right). \end{array}$$

Though the CMNF filter for a nonlinear stochastic system in a rather general form, Equation (Equation 18), which indeed covers the AUV navigation problem at hand, is fully defined by Equation (Equation 6), there are still two questions which need to be discussed in order to clarify all the aspects of the CMNF application on practice: The functions α(x,u), β(x,y), and the calculation of the covariances in Equation (Equation 22).

The basic prediction and correction functions α(x,u), β(x,y) define the structure of the CMNF filter. Their choice is model-specific and it can reflect particular features of the nonlinear functions $\Phi_{k}\left(\cdot ,\cdot \right)$, $\Psi_{k}\left(\cdot \right)$. The common option nevertheless is the prediction “by virtue of the system” and the correction in the form of the residual, which in the case of Systems (Equation 2) and (Equation 17) with $E\left\{W_{k}\right\}=0$ and $E\left\{E_{k}\right\}=0$ are as follows:$$\begin{array}{c} \alpha_{k+1}\left({\hat{X}}_{k},u_{k}\right)=\Phi_{k}\left({\hat{X}}_{k},u_{k}\right)={\hat{X}}_{k}+V_{k}\left(u_{k}\right)\Delta t,\\ \beta_{k}\left({\tilde{X}}_{k},Y_{k}\right)=Y_{k}-\Psi_{k}\left({\tilde{X}}_{k}\right). \end{array}$$

Unlike the EKF and pseudo-measurements filtering, the CMNF approach provides a natural way of data fusion from the INS and external measurements: The basic prediction function $\alpha_{k+1}\left({\hat{X}}_{k},u_{k}\right)$ can be chosen in the form of the estimate from the internal navigation System (Equation 16). Finally, the CMNF estimate, which is based on the data fusion from the dead reckoning seabed sensing and external bearing-only measurements is defined by Equation (Equation 22) with structure functions
$$\begin{array}{c} \alpha_{k+1}\left({\hat{X}}_{k},u_{k}\right)={\hat{X}}_{k}+\Delta {\hat{X}}_{k+1},\\ \beta_{k}\left({\tilde{X}}_{k},Y_{k}\right)=Y_{k}-\left(\begin{array}{c} \frac{Y_{B}-{\tilde{Y}}_{k}}{X_{B}-{\tilde{X}}_{k}}\\ \frac{Z_{B}-{\tilde{Z}}_{k}}{\sqrt{{\left(X_{B}-{\tilde{X}}_{k}\right)}^{2}+{\left(Y_{B}-{\tilde{Y}}_{k}\right)}^{2}}} \end{array}\right), \end{array}$$
where the shift $\Delta {\hat{X}}_{k+1}$ is defined by Equation (Equation 15) and the corresponding algorithm from the Section 4.

The only question left is the covariance matrices, which are necessary for the calculation of the linear estimator coefficients in Equation (Equation 22). The general CMNF approach implies that instead of the real covariances one uses their estimates, obtained by means of the Monte-Carlo sampling.

## 6. AUV Control Simulation

In this section, we provide the results of the numerical simulation of the described filtering and control algorithms. We compare the results of the simulation in a case when the nominal AUV speed $V_{k}$ in the motion dynamic Equation (Equation 2) is known exactly and when it is evaluated by an external estimator—seabed sensing algorithm from Section 4.

The nominal trajectory, i.e., the path, which AUV has to follow, is given by the Equation (Equation 3) with ${\overset{˚}{\gamma }}_{k}=\frac{\pi }{100}\cos\left(\frac{\pi }{100}t_{k}\right)$, ${\overset{˚}{\theta }}_{k}=\frac{\pi }{3}\cos\left(\frac{\pi }{25}t_{k}\right)$, where k=0,…,N, $t_{0}=0$, $t_{N}=T=300$ and $\Delta t=t_{k+1}-t_{k}=1$ second. The absolute value of the AUV speed is constant: ${\overset{˚}{V}}_{k}=2\;m/s$ for all *k* and the initial condition is ${\overset{˚}{X}}_{0}={\left(0,0,-10\right)}^{T}$ (hereinafter all distances are measured in meters). The nominal path of the AUV is shown in Figure 5 and Figure 6 in 3D and component-wise respectively.

It should be noted that this particular path is chosen for modeling purposes only: There was no aim to reflect any specific real-world AUV mission, but simply to get the variation of the components in different scale: ${\overset{˚}{X}}_{k}∼100$ m, ${\overset{˚}{Y}}_{k}∼10$ m, ${\overset{˚}{Z}}_{k}∼1$ m.

The sample paths of the system are simulated using the Equation (Equation 2), where the disturbances $W_{k}$ are i.i.d. Gaussian vectors with zero mean and covariance $S^{W}$ equal to the identity matrix $I^{3\times 3}$: $W_{k}∼N\left(0,I^{3\times 3}\right)$. The initial condition is supposed to have the same distribution: $X_{0}∼N\left(0,I^{3\times 3}\right)$.

In this simulation, it is assumed that on the sea surface there is a set of four sources of acoustic signals (beacons) with known coordinates $X_{B}^{i}$, i=1,…,4: $X_{B}^{1}={\left(500,100,0\right)}^{T}$, $X_{B}^{2}={\left(500,-100,0\right)}^{T}$, $X_{B}^{3}={\left(-100,100,0\right)}^{T}$, $X_{B}^{4}={\left(-100,-100,0\right)}^{T}$. These coordinates are chosen so that the situation of the bearing angles $\phi_{k}$, $\lambda_{k}$ close to $\pm \frac{\pi }{2}$ would not be possible and there would be no need in the regularization of the observations in Equation (Equation 17). The beacons’ positions with respect to the AUV nominal path are shown in Figure 5. Note that despite the fact that in all the previous considerations a single beacon was assumed, the transition to the multiple observations case is rather obvious and requires only vertical stacking of correspondent values in the original Equations (Equation 17) and (Equation 18) and pseudo-measurements of Equations (Equation 19) and (Equation 20).

The precision of the bearing angles measurements is assumed to be equal to 0.5°, which in this simulation means that the noise in Equation (Equation 17) is Gaussian with the distribution $E_{k}={\left(\varepsilon_{k}^{\varphi },\varepsilon_{k}^{\lambda }\right)}^{T}∼N\left(0,\tan^{2}\left(0.5^{\circ }\right)I^{2\times 2}\right)$.

The control sequence is defined by System (Equation 7) with position estimates ${\hat{X}}_{k}^{CMNF}$, ${\hat{X}}_{k}^{UPMF}$ and ${\hat{X}}_{k}^{EKF}$ given by:CMNF: Conditionally minimax nonlinear Filter (Equation 22) with basic structure Functions (Equation 24) and covariances calculated by the Monte-Carlo sampling on a separate test set of $10^{5}$ AUV paths;UPMF: Unbiased pseudo-measurements Filter (Equation 21);EKF: Extended Kalman filter, which provides a common approach to deal with the non-linearity in the system dynamics or observations [18].

The estimates and the correspondent random processes of state evolution (and their samples) are further referenced as ${\hat{X}}_{k}^{F}$ and $X_{k}^{F}=X_{k}\left(u_{k}^{*}\left({\hat{X}}_{k}^{F}\right)\right)$ respectively, where F∈{CMNF,UPMF,EKF}.

The simulation is done for two cases of the prediction ${\tilde{X}}_{k+1}$ used by all three filtering algorithms:prediction by virtue of System (Equation 18): ${\tilde{X}}_{k+1}=\Phi_{k}\left({\hat{X}}_{k},u_{k}^{*}\left({\hat{X}}_{k}\right)\right)={\hat{X}}_{k}+V_{k}\left(u_{k}^{*}\left({\hat{X}}_{k}\right)\right)\Delta t$prediction based on the acoustic measurements: ${\tilde{X}}_{k+1}={\hat{X}}_{k}+\Delta {\hat{X}}_{k+1}$, with $\Delta {\hat{X}}_{k+1}$ defined in Equation (Equation 15).

The former takes advantage of the exact values of the AUV speed, while the latter uses an estimate of the AUV position shift.

For the position estimation with seabed sensing simulation, the following setting is used. The direction of the i,k-th measurement beam from Equation (Equation 8) uniformly varies in ranges $\gamma_{i}\in \left(42^{\circ },82^{\circ }\right)$, $\theta_{k}\in \left(-10^{\circ },30^{\circ }\right)$, i,k=1,…,8. The accuracy of the range measurement is assumed to be 0.1 m, which corresponds to the sonar’s accuracy for the depths of ∼10 m. The unknown seabed profile is defined by the equation ψ(x)=0, where ψ(x) is the following smooth harmonic function:(24)$$\psi \left(x\right)=a_{0}+z+\sum_{l=1}^{K}\left(a_{l}\sin\left(\frac{2\pi l}{P_{x}}x\right)+b_{l}\cos\left(\frac{2\pi l}{P_{x}}x\right)+c_{l}\sin\left(\frac{2\pi l}{P_{y}}y\right)+d_{l}\cos\left(\frac{2\pi l}{P_{y}}y\right)\right),$$
where K=2, the parameters $a_{0}=30.0$ m, $P_{x}=P_{y}=20.0$ are fixed, and the coefficients $a_{l}$, $b_{l}$, $c_{l}$ and $d_{l}$ are randomly generated for each trajectory using the standard uniform distribution.

In Figure 7, we show sample paths of the system for three mentioned state estimators and the nominal trajectory which they intended to follow. In Figure 8, we show the estimate error mean and standard deviation calculated on the set of $10^{5}$ sample paths.

The simulation shows almost equal performance of all three filters. This can be explained by the fact that the chosen acoustic sources positions provide the observations close to linear and in this case all the filters at hand act like the Kalman filter, which is optimal for linear systems.

This, nevertheless, can not serve as a justification for the EKF supreme applicability. In Figure 9, we present the results for the same setting, but with one beacon moved closer to the AUV starting position $X_{0}$, namely $X_{B}^{4}={\left(-5,-5,0\right)}^{T}$ and for a shorter time period equal to 10 s. In this situation the linearization errors play more significant role and hence the EKF demonstrates divergence, while the other two filters remain stable.

In the final simulation we aim to demonstrate how inaccurate description of the system dynamic can dramatically affect the quality of the estimation and hence the control of the UAV. To that end we apply the same three filters used in the previous simulations but now with inaccurate prediction provided by the acoustic measurements. In Figure 10 we show sample paths for the acoustic seabed sensing predictor. The control based on the extended Kalman filter and the pseudo-measurements filter fail to follow the nominal path. In Figure 11, we show the results for the conditionally minimax nonlinear filter based on the data fusion from the dead reckoning seabed sensing and external bearing-only measurements defined by Equation (Equation 22) with structure Functions (24). The increase of the sampled standard deviation shown in Figure 11 in comparison with the prediction by virtue of the system case shown in Figure 8 is not very high. It should be noted, that according to the simulation CMNF exhibits a regular bias. This bias is inherited from the acoustic seabed shift estimate which is also known to have a bias conforming to the nominal path [16].

As a final assessment of the estimate/control performance we provide the sample values of the uniform version of Criterion (Equation 4):$${\bar{J}}^{F}=\max_{k=0,\ldots ,N}J\left(u_{k}^{*}\left({\hat{X}}_{k}^{F}\right)\right)=\max_{k=0,\ldots ,N}E\left\{{∥{\overset{˚}{X}}_{k}-X_{k}^{F}∥}^{2}\right\}$$

The sampling is performed on the same sets of paths as for the calculation of the sample mean and standard deviation presented in Figure 8 and Figure 11. The uniform performance criterion sample values are presented in Table 1. The first row corresponds to the position prediction by virtue of System (Equation 18) and the second row shows the results with prediction obtained by velocity estimation via seabed sensing, Equation (Equation 15). The columns correspond to the filtering algorithms applied: The conditionally minimax nonlinear Filter (Equation 22) with basic structure Functions (Equation 24), the unbiased pseudo-measurements filter, Equation (Equation 21), and the extended Kalman filter. Note that for the seabed sensing prediction only the CMNF filter allowed to acheive a reasonable estimate/control performance.

The simulation shows that in the ideal situation with good initial accuracy and observation conditions close to that of a linear system, the gain in the estimation quality and hence in the control performance is insignificant in comparison with the standard extended Kalman filter. Nevertheless, the proposed filtering algorithms can demonstrate better qualities in less favorable settings, e.g., when one of the beacons is close to the initial point of the path, the EKF diverges, while the proposed filters remain stable. Another result is that the linear filters are highly sensitive to the dynamic model description, and inaccurate parameters’ determination or evaluation can even lead to divergence.

## 7. Conclusions

In the present paper a mathematical model for an AUV navigation system based on the locally optimal (predefined path following) control and position estimation provided by seabed acoustic sensing and external DOA measurements. The performance of the proposed algorithms, namely the conditionally minimax nonlinear filter and pseudo-measurements filter, was evaluated by numerical experiments. It was demonstrated, that the CMNF approach provides a natural way for data fusion from different sources of observation by allowing a certain level of liberty in the basic prediction function choice and, at the same time, it allows the errors of the prediction step to be taken into account. The nonlinear character of measurement equations does not permit to evaluate the control quality in advance and needs a detailed analysis of the sensor fusion, which requires detailed modeling in the settings, which take into account the environment features and noise characteristics. We suppose that the methods considered in the present paper permit one to obtain sufficiently reliable approaches to underwater navigation without costly field experiments.

## Figures and Tables

**Figure 1 sensors-19-05520-f001:**
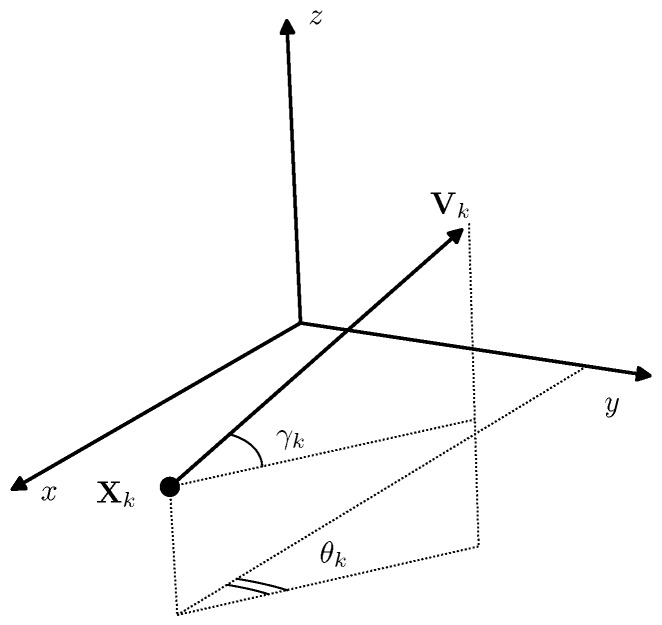
Control parameters $u_{k}=\left(\gamma_{k},\theta_{k},V_{k}\right)$.

**Figure 2 sensors-19-05520-f002:**
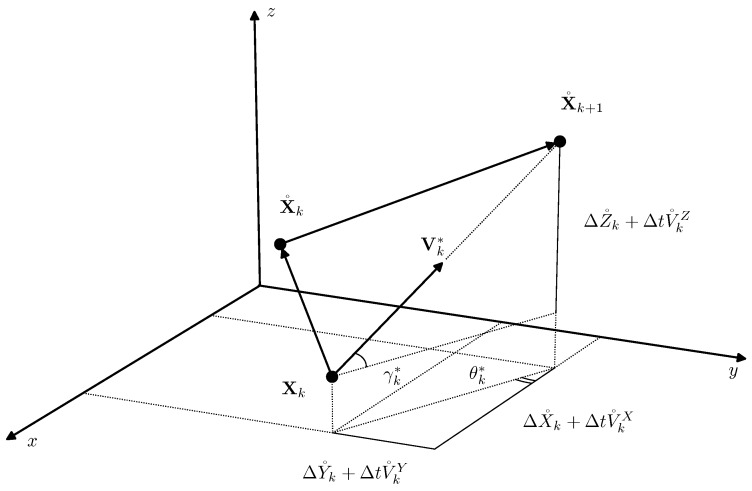
Optimal control geometrical interpretation.

**Figure 3 sensors-19-05520-f003:**
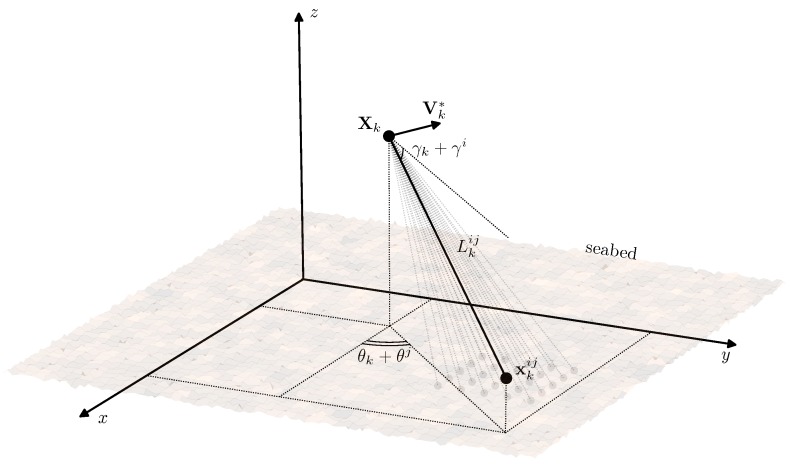
Acoustic beam (i,j) reaching the seabed surface at $x_{k}^{ij}$.

**Figure 4 sensors-19-05520-f004:**
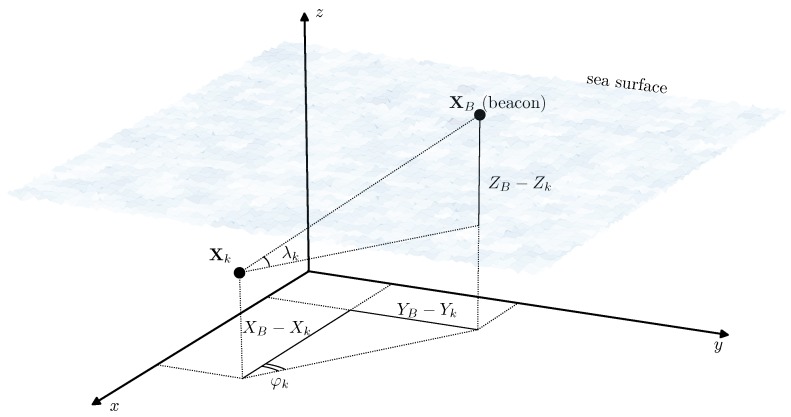
Bearing to the point (beacon) with known coordinates $X_{B}$.

**Figure 5 sensors-19-05520-f005:**
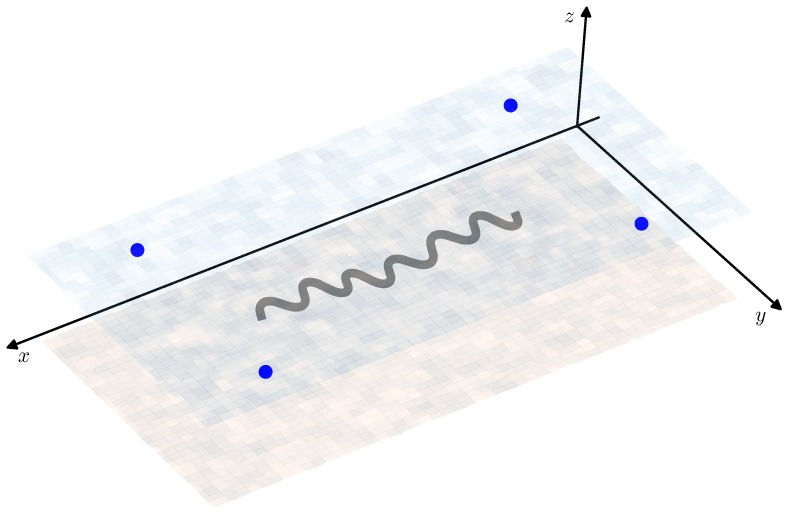
Nominal autonomous underwater vehicle (AUV) path ${\overset{˚}{X}}_{k}$ (grey line) and the positions of the beacons (blue dots).

**Figure 6 sensors-19-05520-f006:**
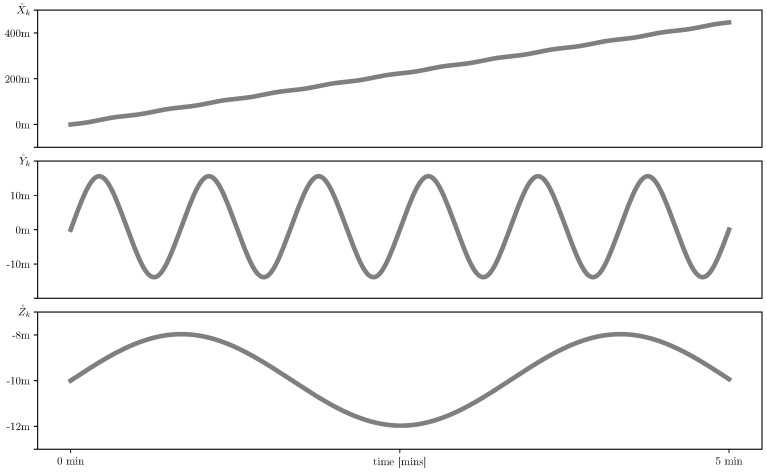
Nominal AUV path ${\overset{˚}{X}}_{k}={\left({\overset{˚}{X}}_{k},{\overset{˚}{Y}}_{k},{\overset{˚}{Z}}_{k}\right)}^{T}$ coordinate-wise.

**Figure 7 sensors-19-05520-f007:**
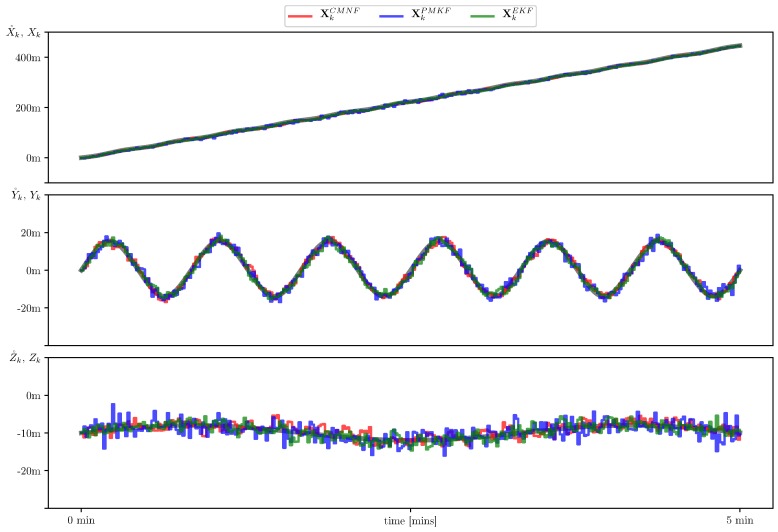
Sample AUV paths with state estimate provided by conditionally minimax nonlinear filtering (CMNF) (**red line**), unbiased pseudo-measurement filter (UPMF) (**blue line**), and extended Kalman filter (EKF) (**green line**) along with the nominal (desired) path ${\overset{˚}{X}}_{k}$ (**grey line**).

**Figure 8 sensors-19-05520-f008:**
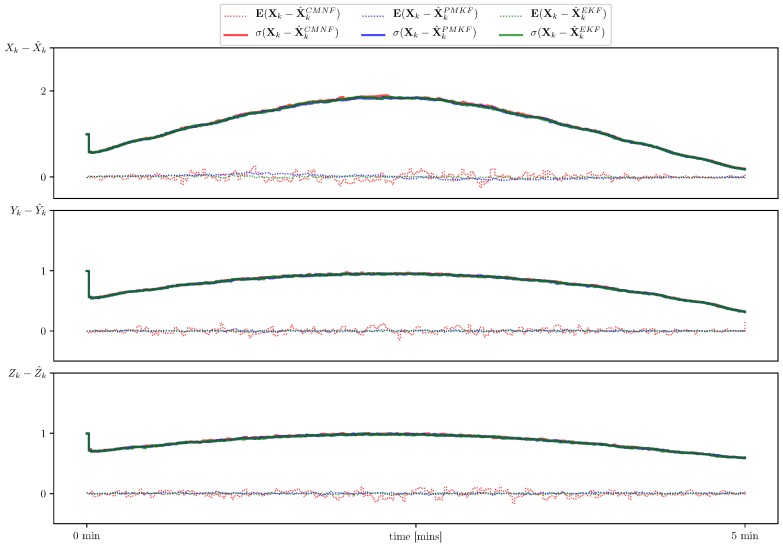
Estimate error sample mean (**dotted line**) and standard deviation (**solid line**) for CMNF (**red**), UPMF (**blue**), and EKF (**green**) estimates.

**Figure 9 sensors-19-05520-f009:**
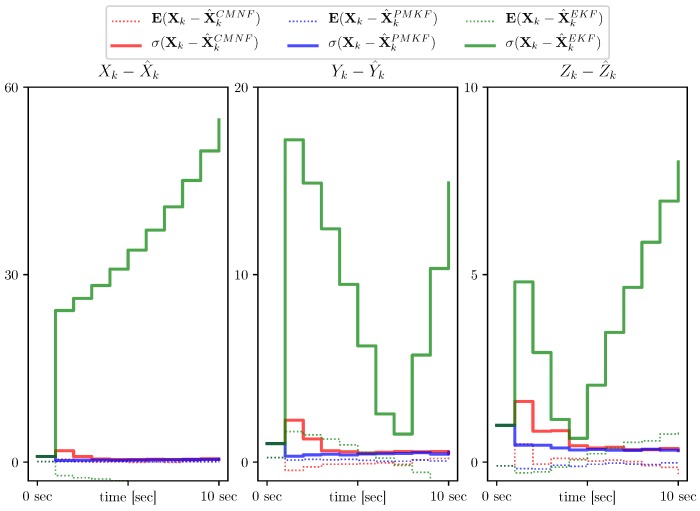
Estimate error sample mean (**dotted line**) and standard deviation (**solid line**) for CMNF (**red**), UPMF (**blue**), and EKF (**green**) estimates for the case of the path close to one of the beacons.

**Figure 10 sensors-19-05520-f010:**
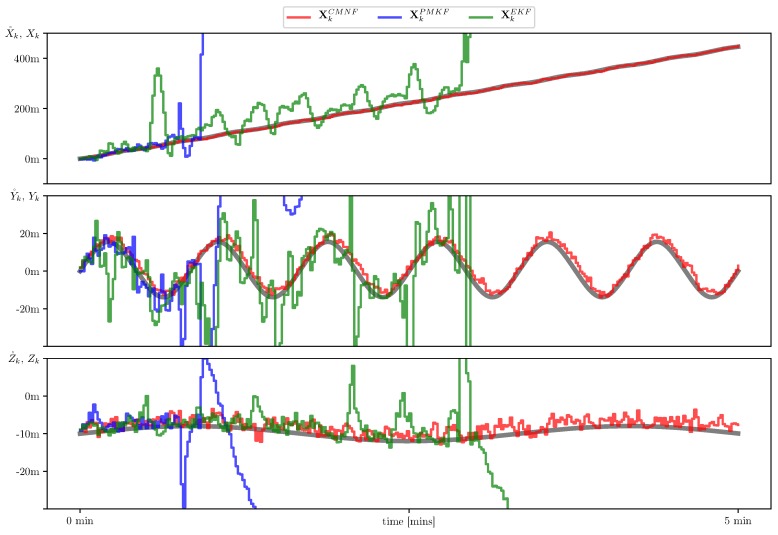
Sample AUV paths with state estimate provided by CMNF (**red line**), UPMF (**blue line**), EKF (**green line**) with prediction based on the acoustic sensing. The nominal (desired) path ${\overset{˚}{X}}_{k}$ (**grey line**).

**Figure 11 sensors-19-05520-f011:**
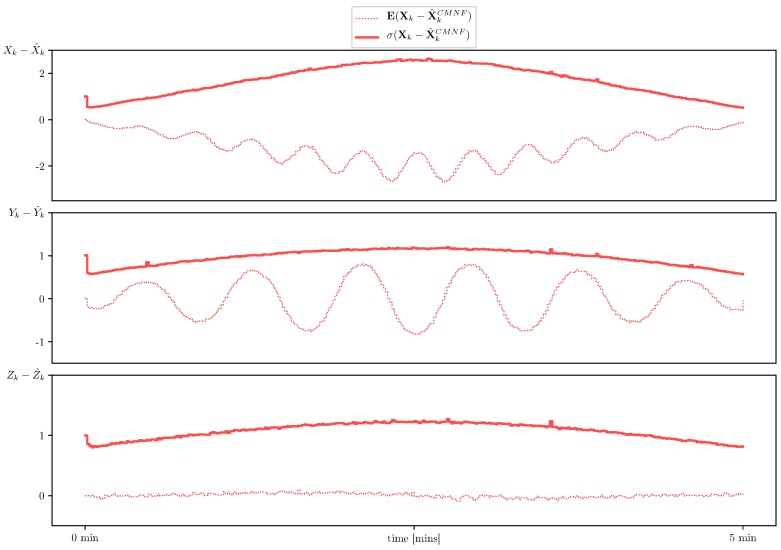
Estimate error sample mean (**dotted line**) and standard deviation (**solid line**) for CMNF (**red**) estimate with prediction based on the acoustic sensing.

**Table 1 sensors-19-05520-t001:** Uniform performance criterion sample values ${\bar{J}}^{F}$, for F∈{CMNF,UPMF,EKF}.

Prediction	CMNF	UPMF	EKF
${\tilde{X}}_{k+1}={\hat{X}}_{k}+V_{k}\left(u_{k}^{*}\left({\hat{X}}_{k}\right)\right)\Delta t$	14.89	14.80	14.93
${\tilde{X}}_{k+1}={\hat{X}}_{k}+\Delta {\hat{X}}_{k+1}$	27.25	—	—

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
