# Peer review of "On AUV Control with the Aid of Position Estimation Algorithms Based on Acoustic Seabed Sensing and DOA Measurements"

_sensors, 2019, doi:10.3390/s19245520_

Round 1

Reviewer 1 Report

In this paper, the authors propose a mathematical model for an AUV navigation system based on the locally-optimal (predefined path following) control and position estimation provided by seabed acoustic sensing and external DOA measurements. The research presented in this paper looks solid and interesting. Some suggestions are presented as follows.

The authors are suggested to summarize all notation and symbols and their meaning. The authors are suggested to compare the proposed model and some common used model in terms of error and etc.

Author Response

We are very grateful to the reviewer for high evaluation of our research and useful suggestions. We've done our best in order to correct our paper in accordance with the reviewers' comments.

In particular:

>The authors are suggested to summarize all notation and symbols and their meaning.

We've added a section after introduction which summarizes the notation used througout the paper.

>The authors are suggested to compare the proposed model and some common used model in terms of error and etc.

Our model of the AUV motion is standard but doesn’t take into account the attitude evolution. However, taking into account the difference
in position for different acoustic sensors might be important for large-sized AUV where such sensors are distributed along the AUV body.
Mathematically it is the same filtering problem which already considered in the article, moreover, parallelization of measurements and the
estimation updates may further increase the estimation accuracy. The measurement model used in the article is also more or less standard
since all of them are based on acoustic measurements of the distance from the source on AUV and the acoustic beam reflector. Meanwhile it
makes sense to underline that we focus our attention to the bearing measurement since the range measurements are based on the knowledge of
sound velocity which needs to take into account the distribution of plenty the environmental factors such as temperature, salinity,
therefore we exploit the range metering for relatively short distances, such as from the AUV to the seabed. Moreover, it is difficult
to agree with the Gaussian distribution of error in measured bearing angles, though it becomes commonplace in all articles related
to bearing-only estimation. We suppose that our model where not the angles but their tangents are measured is more realistic and don’t
need the knowledge of the error distributions. As for the comparison in terms of error, we provide it in the numerical simulations section:
the proposed filtering techniques are compared with the extended Kalman filter, which is the most common instrument in dealing with
nonlinear stochastic dynamic models.

Reviewer 2 Report

Please to see the attached file.

Thank you.

Author Response

We are very grateful to the reviewer for high evaluation of our research and useful suggestions. We've done our best in order to correct our paper in accordance with the reviewers' comments.

In particular:

>Errata detected (grammar, lexical, typos,…)
Line 305: It must be some error when fixing =2 , because in the expression for ?(?) , ? is the index for the summation. Perhaps authors wanted to say ?=2?

This error was corrected.

>Formatting failures
Page 13: The expression for ?(?) defining the seabed profile must be labelled as the rest of the equations in the manuscript: “(x)” , where x = equation number.

The expression now has a number.

Reviewer 3 Report

The article discusses various approaches to the control of autonomous underwater vehicle (AUV) with the aid of different velocity-position estimation algorithms. The ideas in the paper are interesting and the theoretic results obtained have some potential in applications. However, in the current version, the manuscript needs minor revisions. My comments are as follows:

(1)The language in this manuscript is good, but it needs to be further polished.

(2)The figures in the manuscript should be further improved, and the curve in the figures should be beautified.

(3)The authors do not give a detailed introduction and description of Table 1.

(4)The current conclusion is too long and needs to be further condensed. Some discussion of the results in the simulation experiment should be demonstrated in Section 6.

Overall, I strongly recommend the authors address these issues and resubmit the manuscript. This is very interesting work and would be of great value to the community.

Author Response

We are very grateful to the reviewer for high evaluation of our research and useful suggestions. We've done our best in order to correct our paper in accordance with the reviewers' comments.

In particular:

>(1)The language in this manuscript is good, but it needs to be further polished.

We've thoroughly examined the paper and corrected all the typos, errors and stylistic issues we were managed to find.

>(2)The figures in the manuscript should be further improved, and the curve in the figures should be beautified.

In order to improve the figures we've done the following steps with the purpose of their beautification (at least how we've understood it):
1. the curves in figs are now presented in a piece-wise deterministic form to reflect the discrete-time nature of the model;
2. the limits in figures 7 and 10 were adjusted;
3. figure 9 was revised and now it better demonstrates the divergence of the EKF.

>(3)The authors do not give a detailed introduction and description of Table 1.

We've added the necessary clarification before Table 1.

>(4)The current conclusion is too long and needs to be further condensed. Some discussion of the results in the simulation experiment should be demonstrated in Section 6.

The conclusion was revised, and the discussion concerning the simulation results was moved to the correspondent section.